# Student Academic and Social Engagement in the Life of the Academy—A Lever for Retention and Persistence in Higher Education

**Maria José Sá** 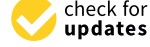

CIPES–Centre for Research in Higher Education Policies, 4450-137 Matosinhos, Portugal; mjsa@cipes.up.pt

**Abstract:** Research studies worldwide have focused on higher education dropout, persistence, and success. Given the profound changes in higher education that have taken place in recent decades, higher education institutions need to compete for students by attracting, retaining, and, ultimately, graduating them. Thus, higher education institutions increasingly offer actions that aim to foster student success. While a smooth and supported process of student transition from secondary to tertiary education is one of the key variables in higher education student retention and paramount for preventing student dropout, the student's overall experience in higher education plays a pivotal role in their performance and success. This paper focuses specifically on higher education students' academic and social involvement, notably through their engagement in extracurricular activities and decision-making processes, which are perceived as critical mechanisms in their persistence in higher education. The study used a qualitative approach with the analysis of four Portuguese higher education institutions. Data were collected through in-depth interviews with students and institutional leaders, complemented with document analysis, and explored through content analysis. The results reveal that, from the wide range of opportunities for involvement offered to students by the higher education institution, activities of an academic nature are the most sought after by students to complement their educational experience. However, students perceive involvement in extracurricular activities in general as critical, both to their overall education and preparation for the labor market and to an easier integration into the institutional environment. Hence, student involvement in cultural or recreational activities, alongside their involvement in institutional decision-making bodies and associative movements, is a privileged way of complementing students' academic training and is perceived by them as important in their overall education, both as professionals and as individuals.

**Keywords:** student experience; student engagement; higher education; extracurricular activities



## 1. Introduction

The major reconfigurations that the Higher Education (HE) arena worldwide has been subject to over the past decades (in particular with the massification of this educational level, the increasing scarcity of resources, especially in financial terms, and the institutional competition for students) have caused a shift in the way higher education institutions (HEIs) relate with their public [1,2]. Thus, the academic community, governmental structures, and HEIs have come to focus on the issues related to how students seek HE and, once in the system, how they integrate, perform, become academically involved, and attain success. In this context, one of the HEIs' major concerns is related to student retention and dropout rates.

The literature, both seminal and more recent, acknowledges the transition from high school to higher education as one of the most complex in students' educational paths [3–16]. This challenging transition may affect their performance and, ultimately, their success in HE [17]. Among HEIs' concerns is, therefore, that this process runs with minimum incidents and as smoothly as possible for first-year students, as this is the year with the

highest dropout rate, in part due to students' difficulties in adapting to the higher education context [13,18].

After the process of transitioning from secondary to tertiary education is completed and students persist, they continue to face many complex challenges throughout their educational path in higher education. The purpose of this study is to get an insight into the factors that, from both the students' and institutional leaders' standpoints, influence their overall experience in HE and which reflect on their path, performance, and academic success. The literature identifies one of these factors as student involvement, also often described as academic and social integration. There is a well-documented direct and positive relationship between students' academic and social involvement as well as their persistence and graduation in the literature [19–22].

According to Astin [11,12,23–25], one of the first researchers to address this topic and the author of a seminal work, students' involvement in the life of the academy plays a paramount role in their retention. His Theory of Student Involvement seeks to explain the dynamics that underlie students' change and development [23]. Using the author's own terms, "students learn by becoming involved" [24] (p. 133).

This paper presents part of the results of a broader study that seeks to obtain a deeper understanding of students' perceptions about their success in HE as a result of their overall experience in the HEI and focuses specifically on students' involvement in extracurricular activities and institutional decision-making processes as a mechanism to enhance their overall experience in HE.

The concept of student involvement in the life of the academy is not a novelty, and the literature offers a wide array of studies addressing this topic [11,12,19–21,23,24,26–32]. According to Astin [12,25], students' involvement consists of the amount of physical and psychological energy they invest in their academic experience; highly involved students devote a considerable amount of their time and energy to their studies, spend a good part of their time on campus, participate in campus activities, and interact with their peers and faculty. Conversely, students with a low degree of involvement neglect their studies, spend very little time on campus, do not participate in extracurricular activities, and have very sporadic contact with peers and faculty. The concept of engagement used by the author involves a behavioral component, as "[...] It is not so much what the individual thinks or feels, but what the individual does, how they behave, that defines and identifies involvement" [12] (p. 519).

Although Astin [11,23,25] ascribes high relevance to the role of the institutional environment by providing students with opportunities and possibilities to engage socially and academically with ideas, people, or activities, the author places the student at the center of this process. The author [9,11,20,22,23] argues that the change will occur to the extent that students capitalize on the opportunities offered to them and take an active attitude of involvement. Thus, students' change and potential development are not only effects of the organizational environment but also mainly a result of the quality of their commitment and involvement in the opportunities offered by the HEI [6,19,26]. According to Astin's [12] Theory of Student Involvement, for a curriculum to reach the expected goals, it should enable students to apply effort and investment in terms of energy so that the desired learning and development can occur, emphasizing their active participation in the learning process. Students' time is envisaged in this theory as the most important institutional resource, as the attainment of developmental goals on the part of the students is a direct consequence of the time and effort they commit to academic activities.

Bergmark and Westman [33] sustain that this engagement may take the form of joint participation, notably with faculty and other stakeholders, in developing partnerships, co-creating curriculum, and acting as agents of educational change. Student engagement, both at the academic and social level, may thus foster their overall development and the feeling of belonging [33]. Moreover, this active institutional and social participation and engagement is also critical in the development of transversal competences by HE students (such as communication, teamwork, research, problem-solving, leadership, creativity,

critical thinking, and time management), which are paramount in personal enhancement and professional preparedness [34,35]. In the same vein, Bergmark and Westman [33] maintain that student participation influences their engagement and motivation to attain the knowledge and competences that will be critical in their future profession.

Based on Astin's Theory of Student Involvement [12,24], Berger and Milem [36] argue that the more involved students are in the life of the HEI, the more likely they are to interact with and be affected by the environment of the campus. Moreover, the campus organizational environment and students' perceptions of that environment directly influence the extent and types of student involvement [36]. This involvement is not limited, naturally, to students' participation in academic activities but far exceeds these limits and is materialized, to a large extent, in their involvement in extracurricular activities of a more social or recreational nature [37].

The literature on this topic acknowledges and emphasizes the relevance of students' social experiences in their integration into the institution and their subsequent commitment to it, e.g., [6,9,19,21,30,36,38–40], among others. Spady [41], in his seminal work, posits that students' integration into the life of the institution depends on their ability to successfully respond to the demands of both the social and the academic systems in which they are placed. The efficacy of this process results, according to the author, in the students' higher satisfaction, which leads, in turn, to a higher level of institutional commitment and, thus, to an increased likelihood that they persist. This assumption is shared by Tinto [8–10], who advocates that the degree of students' interaction with the academic and social environment of the HEI determines their conduct in terms of persisting in or dropping out of the institution. In line with Tinto [8,9], Braxton et al. [42] argue that the higher the level of social integration, the higher the subsequent level of commitment to the institution.

Acknowledging the central importance of HE students' involvement in their academic experience, Kahu [29] offers a model that, like Astin's [12] Theory of Student Involvement, places the student in the center of the teaching and learning process, pointing out the complexity of factors influencing their involvement, and perceiving students' involvement as a psychosocial process, influenced by institutional and personal factors. Among them, the author highlights the students' relationships with peers, faculty, and non-teaching staff; motivation; and the students' personalities, among many other factors [29]. Building on this model, Kahu and Nelson [43] emphasize that an engaged student better attains not only scientific knowledge and competences but also academic success and personal development. Furthermore, the authors upgrade Kahu's [29] model by analyzing student transitions. They conclude that the interactions between the students' individual traits and the institutional factors are critical for student engagement, highlighting the importance of a close relationship between the HEI and its students. Furthermore, these personal and institutional factors do not function separately, and there is a need for interaction between them to enhance student engagement [43].

Also based on Astin's [24] Theory of Student Involvement and on its prerogative that students' academic and social involvement plays a central role in shaping their outcomes, Berger and Milem [36] argue that the more involved students are in the life of the academy, the more likely they are to interact with and be affected by the environment of the campus. Still, according to the authors, the organizational environment of the campus and students' perceptions of that environment have a direct influence on the extent and types of students' involvement in academic and social activities. The more positive the students' perception of the organizational environment, the greater their likelihood of participating in various academic and social activities promoted and made available by HEIs and, at the same time, persisting in the institution. Subsequent studies have achieved similar conclusions, thus showing the centrality of the relationship between the institutional environment and students' engagement in different kinds of activities, as well as the fact that the student experience is enhanced with their participation in academic and social activities inside and outside the classroom, e.g., [44–48].

Moreover, student participation and engagement in institutional decision-making bodies may also be a booster for a fulfilling overall experience of HE students. The activities that allow for this involvement encompass rather simple ones, such as informal strategies inside and outside the classroom, and more structured and broadened ones, such as the active participation of students in "institution-level systems for student representation" [49] (p. 688), where they may be heard at the formal institutional level and be effective actors in the institutional decision-making process. Pascarella and Terenzini [6] argue that if students' involvement and commitment are key pieces of this puzzle, then HEIs should focus on ways and mechanisms they can use to keep their students involved and committed. According to the authors, this is possible by promoting academic, social, and extracurricular dynamics and offers that might foster students' involvement and commitment to their individual goals and the HEI as a whole. On the other hand, students' involvement in the life of the academy is potentially influenced by the way they experience the HEI's organizational attributes. Organizational attributes such as participation in organizational decision-making, justice in policy administration, and communication may affect students' decisions to persist in or drop out of the institution [50].

Considering the above, it is safe to ascertain that student engagement is a variable with relevant weight in the definition of higher education student success [51,52]. Furthermore, student engagement and participation influence student retention [53]. The study aimed to assess the degree of importance of extracurricular activities in higher education students' integration and adaptation to the HEI as well as the role they ascribe to these activities in their overall development. Given the goal of the study, the following research questions guided the research:

1. How do students mobilize and get involved in extracurricular activities, and to what extent do these actors understand this involvement as relevant to their academic experience?
2. How do students and HEIs view their participation in institutional decision-making?

## 2. Materials and Methods

### 2.1. Research Design

To obtain the perceptions of the institutional actors involved and, therefore, meet the goal of the study mentioned above, the methodology chosen for this research study is qualitative, and the multiple case study methodology was used. As argued by Baxter and Jack [54], this methodological approach enables the exploration of a phenomenon in the context in which it occurs, using a wide range of data collection sources. Moreover, qualitative methods are better suited to grasping and interpreting the meanings [55–57], that is, specifically in this research study, how students perceive and represent their academic experience in terms of integration in the HEI. The qualitative approach allows for obtaining a thorough and detailed understanding of the phenomena studied in their own context and based on the perceptions of the actors involved in them [56].

### 2.2. Participants

The sample of this study is composed of 58 academic actors, including students and institutional leaders, from four Portuguese HEIs (two universities and two polytechnic institutes, all from the public subsystem). Concerning students, the study sample is composed of 40 HE students. For the selection of participants from the four HEIs, the following criteria were considered: the nature of the study program (1st and 2nd cycle, or degree and master's degree); the nature of knowledge (hard-pure, soft-pure, hard-applied e soft-applied [58]); and gender. Of the 40 participant students, 21 (52.5%) are undergraduate students, and 19 (47.5%) are graduate students. The participants are 19 male and 21 female students. The analysis of social-economic-educational indicators reveals the prevalence of students coming from middle-class families and whose parents have compulsory education qualifications (in Portugal, the 12th grade).

As for the institutional leaders, the sample is composed of 18 institutional actors that have some level of influence on the institutional decision-making process (e.g., Vice/Pro-Rectors and Vice-Presidents that deal with issues concerning quality, training, education, organization, and academic activities). Moreover, the Student Ombudsman, students' representatives on the Pedagogical Council, and presidents of Student Unions are also part of the sample. The sample selection sought to reflect the heterogeneity of the wider group by following the sample section criteria mentioned above.

Contrary to the sample of institutional leaders, the sample of students was not defined a priori. According to the literature [59,60], it is common for the size of the sample to take shape as the study progresses until no new categories, themes, or explanations emerge from the data, at which point theoretical and empirical saturation is reached. In the study, data saturation was reached with the 40th interview. The strategy adopted in the design of the sample consisted of a sampling procedure according to which the researcher has access to informants through other informants, which gives rise to a "snowball" sample [61,62]. The selection of undergraduate and postgraduate students aimed to obtain the perceptions of both groups regarding the integration processes and the involvement in extracurricular activities, as students from each group have distinct features and hence may perceive these processes differently.

The mean age of the students is 23.28 (σ 2.82), with the maximum age being 35 and the minimum age being 19. Regarding the distribution of the sample by age range, more than half of the students in the sample (60%) are between 21 and 25 years old; 22% of the students are between 18 and 20 years old; 10% of students are between 31 and 35 years old; and 8% are between 26 and 30 years old.

### 2.3. Data Collection Tools

Data collection tools are threefold: (1) semi-structured interviews with students; (2) semi-structured interviews with institutional leaders; and (3) document analysis to allow for data triangulation. Tables 1 and 2 detail the questions of the semi-structured interview with students and institutional leaders that specifically concern student integration, adaptation, and engagement, which were generated based on the literature review.

**Table 1.** Questions from the semi-structured interview with students.

| Dimensions | Questions | |
|---|---|---|
| | **1st Cycle (Bachelor's)** | **2nd Cycle (Master's)** |
| Integration/adaptation to the HEI | • Do you like the HEI you enrolled in? <br> • How would you describe the environment surrounding the HEI? <br> • How do you consider the opening hours and operation of the services? <br> • How was your experience regarding hazing? Do you think that it helped you integrate into the HEI? | • Do you like the HEI you enrolled in? <br> • How would you describe the environment surrounding the HEI? <br> • How do you consider the opening hours and operation of the services? |
| Adaptation to the study program | • Do you like the program you are enrolled in? <br> • Do you believe it is well structured/organized and its subjects have quality and relevance? <br> • Do you believe there is a link between curriculum content and professional opportunities? <br> • How much of your time and energy do you devote, on average, to your studies? <br> • In terms of time management, do you believe that the workload allows you to prepare yourself in academic terms? | |
| Involvement in extracurricular activities | • Are you a part of any academic association? <br> • Do you get involved in cultural, recreational and/or sporting activities at the HEI? | |
| Relationship with peers | • How would you describe your relationship with your peers in academic and social terms (i.e., inside and outside the classroom)? | |
| Relationship with teachers | • Do you think there is a good relationship between teachers and students? <br> • Do you believe contact with teachers inside and outside the classroom is easy? <br> • Do you believe teachers show the availability of time to support/interact with students? | |
| Attainment of educational goals | • Do you expect, with your overall experience in HE, to attain the goals you set when you came here? | • Do you expect, upon the conclusion of the master's, to attain the goals you set when you came here? |

**Table 2.** Questions of the semi-structured interview with institutional leaders.

| Interviewees | Dimensions | Questions |
|---|---|---|
| Heads of HEIs | Internal structures, policies and practices | • What is the HEI's vision of the student?<br>• How is institutional communication carried out with students on subjects of interest to them (they knowledge they have about academic and social rules, etc.)?<br>• What is the level of student participation in organizational decision-making on the development of social and academic rules?<br>• What actions are taken to listen to students about their experience at the HEI?<br>• In terms of the physical structure of the institution, how do you assess the quality of the facilities (in terms of curricular and extracurricular activities)?<br>• What actions are carried out by the HEI to integrate new students? |
| | Curricular and co-curricular programs, policies and practices | • Regarding students' involvement in academic life, whether in academic or social terms, what are the opportunities, stimuli, and possibilities for this involvement to be effective? |
| Student Ombudsman | Student support | • What kind of problems do students bring with them when they come to you?<br>• In what ways does this body address these requests?<br>• What type of students seek the Ombudsman's help the most?<br>• How often do students use this support? |
| | Integration of new students | • Are there any initiatives carried out by this body to welcome and integrate new students?<br>• What is, specifically and in practice, the support provided by the Student Ombudsman to new students? |
| | Organizational structure | • How would you characterize the institution in terms of (a) study spaces/classes and (b) leisure spaces?<br>• How do you characterize the environment on campus?<br>• How do you envision student participation in institutional decision-making? |
| | Relationship with the HEI | • How do you characterize the relationships that students establish in this institution in terms of (a) the relationship between peers, whether in academic or social terms; (b) the relationship with teachers, inside and outside the classroom; and (c) the relationship with the various services and non-teaching staff? |
| President of the Student Union (SU) | Student support | • What are the motivations for students to join the SU?<br>• What kind of problems do students bring when they seek the SU?<br>• How does the SU address these requests?<br>• What kind of students are most interested in the SU?<br>• In what academic year are the students enrolled who most seek SU for support in solving problems? |
| | Integration of new students | • What initiatives does the SU carry out to welcome and integrate new students?<br>• What is the support provided by the SU to new students? |
| | Curricular and extracurricular activities | • What activities does the SU promote to support students on their academic path?<br>• What kind of extracurricular activities does the SU carry out?<br>• What is the participation of students in these activities? |
| | Organizational structure | • How would you characterize the institution in terms of (a) study spaces/classes and (b) leisure spaces?<br>• How do you characterize the environment on campus?<br>• How do you envision student participation in institutional decision-making? |
| | Relationship with the HEI | • How do you characterize the relationships that students establish in this institution in terms of (a) the relationship between peers, whether in academic or social terms; (b) the relationship with teachers, inside and outside the classroom; and (c) the relationship with the various services and non-teaching staff? |
| Student Representative in the Pedagogical Board (PB) | Role and goals | • What is the role of the student representative with the PB?<br>• In personal terms, how do you see your role in the PB? |
| | Representingstudents | • What motivated you to apply for this position?<br>• What are your skills and powers as a student representative with the PB?<br>• What are the advantages for students of having their representative at the PB?<br>• What problems/issues do you, as a student representative, carry to PB meetings?<br>• Is there feedback from you to the students you represent? If so, how is this feedback provided? |
| | Relationship with the HEI | • How do you characterize the relationships students establish in this institution in terms of their (a) relationship with the HEI in general, the various services, and non-teaching staff; and (b) academic or social relationships between peers? |

*2.4. Procedure*

One of the data collection sources used was document analysis, which consists of "[ . . . ] a systematic procedure for reviewing or evaluating documents—both printed and electronic" [63] (p. 27). This tool allowed the gathering of information that enables data triangulation, thus increasing the trustworthiness of the study [64]. In this study, documents made available by the HEIs were analyzed, namely their strategic plan, activities report, and other documents, as well as the HEIs' website on the Internet. The second source consisted of in-depth interviews with students and institutional leaders, which took place on the premises of the HEIs analyzed. The interviews started by providing the participants with information on the goals of the study. Subsequently, respecting the ethical principles of the research, the interviewees were assured of the anonymity and confidentiality of the

data they would convey [65]. The interviews were audio-recorded after authorization from the interviewees and transcribed in full for later analysis.

*2.5. Data Analysis*

The research technique used in the analysis of the collected data consists of content analysis, which enables the compression of a large amount of words and text into fewer content categories based on explicit coding rules [66]. In this study, according to the methodological approach used, a set of categories coded a priori have been built based on the critical review of literature carried out on the dimensions and factors affecting HE students' academic and social experiences. However, the categorization initially created was not intended to be closed; data analysis allowed the establishment of emerging categories, i.e., categories that, according to their relevance to the study, have been incorporated into the categorical system throughout the data analysis. The data were coded by paragraph and sentence, as proposed by Strauss and Corbin [67]. Table 3 depicts the themes, categories, and sub-categories that form the categorical system that deals specifically with student integration into and engagement with the life of the academy.

**Table 3.** Themes, categories, and sub-categories of the study.

| Dimension | Theme | Categories | Sub-Categories |
|---|---|---|---|
| Individual | | Integration | • Integration/adaptation to the HEI<br>• Integration/adaptation to HE |
| Organizational | Internal structures, policies and practices | Vision of the student | • Student follow-up practices<br>• Participation of students in institutional decision-making<br>• Student support mechanisms<br>• Quality of the physical structure for (a) academic activities and (b) social/leisure activities |
| Academic Experience | Interpersonal | Curricular and co-curricular programs, policies and practices | • Opportunities, incentives and possibilities for academic and social involvement |
| | | Relationship/interaction with teachers and peers in the classroom | |
| Social Experience | Interpersonal | Relationship/interaction with teachers and peers | • Affective dimension<br>• Instrumental dimension<br>• Limited to the classroom<br>• Beyond the classroom<br>• Depending on the teachers |
| | | Involvement | • Academic activities<br>• Cultural activities<br>• Leisure/sports activities<br>• Non-involvement |
| | | Social integration/adaptation | • Integration by peers<br>• Integration by the HEI |
| Functional Experience | | Operational interaction | • Bureaucratic processes and procedures<br>• Student suitability |
| | | On-campus spatial orientation | • Spatial orientation within the campus |
| | | Institutional Environment | • Human dimension<br>• Physical dimension |
| | | Relationship/interaction with non-teaching staff | • Affective level of interactions<br>• Functional level of interactions |

Following Saldaña [68], data coding began when the first interviews and institutional documents were analyzed in the pre-coding phase. In the second phase (or first coding cycle), the codes were refined and organized into categories and sub-categories through a deeper analysis of the data. Finally, in the third phase (or second coding cycle), the data were compared and consolidated.

## 3. Findings

Students' involvement and commitment are pivotal in their academic and social developmental processes [12,19,26,36,48], and HEIs have part of the responsibility for this process. This responsibility has to do with promoting dynamics and the production of offers at the academic, social, and extracurricular levels that can promote such involvement

and commitment, both with students' personal goals and with the HEI [6,50]. Considering the learning process as global and comprehensive, HEIs should rethink the organizational structures so that it is possible to incorporate and capitalize on the students' experiences that occur outside the classroom, trying to connect the formal and informal learning contexts, and taking themselves as organizations focused on learning rather than on instruction [6].

The models that analyze HE students' experiences argue that their participation in extracurricular activities plays a relevant role, on the one hand, in their involvement with the institution and, on the other hand, in their willingness to seek the achievement of their personal goals and targets. This phenomenon has clear consequences in terms of the students' likelihood to persist in or drop out of their studies in a given institution or even in HE [6,12,19–21,26,27,31,36,69,70].

The analysis of the students' narratives regarding their involvement in extracurricular activities allowed the identification of four sub-categories: (1) involvement in academic activities; (2) involvement in cultural activities; (3) involvement in recreational/sports activities; and (4) involvement in associative activities. Moreover, the institutional perspective on this topic is also analyzed.

*3.1. The Students' Perspective*

3.1.1. Involvement in Academic Activities

Academic events are the most frequently mentioned ones by respondents and those that come first on the list of their preferences in terms of involvement in extracurricular activities. Students state that whenever they have the opportunity, they participate in lectures, workshops, conferences, and seminars, especially if they fall within the scope of their educational field, as they perceive these moments as relevant contributions to their academic preparation and overall education. These students perceive that these educational moments may be a lever for networking, for closer contact with their future professional area, and for a first approach to the labor market and the opportunities therein. One student states: "I regularly participate in events of an entrepreneurial nature because I know I'll have to work for a boss at first, but I do not see myself doing that for the rest of my life, and I know that I have to take the initiative and get the most enjoyment from it" (undergraduate student). This stance reflects a logic of preparation to attain personal and professional goals in the future. Students see these events as opportunities to grasp what is happening in the labor market insofar as "[ . . . ] people who come here to speak in these lectures are either starting a business or already run one and come here to share their experience with us, which is always an added-value" (undergraduate student).

3.1.2. Involvement in Cultural Activities

The second group of activities preferred by the students surveyed regards cultural events, such as plays, music concerts, and cinema. They perceive these activities as also educational opportunities, but mainly as moments where they can interact with others and strengthen their social relationships outside the classroom. As a graduate student states, "I participate in some cultural activities because I think it's good to foster team spirit [ . . . ] I think it brings people together". However, it is stressed rather frequently that, depending on time management and the availability of financial resources, these activities lose out to the more academic ones, such as lectures and workshops, for example.

The analysis of the perceptions of the institutional leaders surveyed regarding students' involvement in cultural activities reveals that they clearly acknowledge the educational value of these activities and, thus, they seek, through the promotion of diversified actions, to engage their students in activities that are "[ . . . ] structured and that develop in students a sense of responsibility and commitment" (Vice-Rector). The institutional acknowledgement of students' involvement in such initiatives is subsequently "[ . . . ] endorsed in the diploma supplement ("The Diploma Supplement is produced by higher education institutions according to standards agreed by the Commission, the Council of Europe and the United Nations Educational, Scientific and Cultural Organization (UNESCO). It is also part of

the Europass framework transparency tools. The Diploma Supplement is designed as an aid to support the recognition of academic qualifications. The Diploma Supplement is an important tool of the European Higher Education Area for graduates to ensure that their degrees are recognized by higher education institutions, public authorities and employers in their home countries and abroad. It does, however, not represent a Curriculum Vitae or a substitute for the original qualification" [71]; it is an activity that the student carried out without receiving any money but that has contributed to his/her education" (Vice-Rector). According to this leader, the endorsement in the Diploma Supplement of activities in which students participated and had active involvement is an institutional validation of such participation as a complementary form of education. It is stated that this "[ . . . ] is a way of saying that we value their engagement in these activities, and also that we believe that this is also education, not in the strict academic sense, but educational activities during the academic period" (Vice-Rector).

### 3.1.3. Involvement in Recreational/Sports Activities

A third group of activities consists of recreational or sports events, which are attractive to some students and in which they participate whenever they have the opportunity to do so. Some of these activities include sports tournaments, football games, dinners, and parties with friends. Similarly to what happens with cultural activities, these social events are perceived as privileged moments to interact and strengthen ties with colleagues and friends. In short, social events of a more recreational nature are perceived by these students not only as moments of relaxation but also as opportunities to strengthen ties of friendship established with peers.

### 3.1.4. Involvement in Associative Activities

For some of the students interviewed, participation in associative activities—student unions, student study groups, and academic choirs, among others—is also perceived as a positive form of active involvement in the life of the academy. It also functions as a way to attain social and professional tools that will be useful for them in the future. On the other hand, the assumption of responsibilities as representatives of their peers in the institution's management bodies is also referred to as relevant to their experience in HE.

For these students, the extracurricular activities made available by the institution play a relevant role in their experience in HE because, as they state, "[ . . . ] there is much more to life than university" (graduate student). They perceive these activities as developmental moments because, according to a student, "[ . . . ] these events add to my education and help me feel integrated into the academic community" (graduate student). These students believe that these moments always provide some kind of learning, and they are also acknowledged as privileged ways of student integration into the academic community.

Although most students interviewed mention that they participate, to a greater or lesser extent, in extracurricular activities as part of their educational process, there is a small group of students who have the opposite behavior—very low adhesion or even non-adhesion. The most commonly cited reasons for this social behavior have to do mostly with financial constraints and time availability, but also because these students do not see their involvement in these activities as a priority, choosing to study instead. Another reason given was the obvious lack of interest in these activities.

From the set of extracurricular activities that the HE students affirm that they engage in, the analysis of the narratives allowed for the identification of the four sub-categories described above. Figure 1 provides an overall view of these four sets of extracurricular activities as well as their relative relevance, as depicted by the different sizes of the circles.

These results are in line with previous studies that reveal the participation of students in extracurricular activities has benefits for student learning and career development [37], as well as the attainment of both technical and transversal skills [33,72]. Furthermore, participation in extracurricular activities has a significant influence on the engagement levels of students when compared to those who do not get involved in these activities [73].

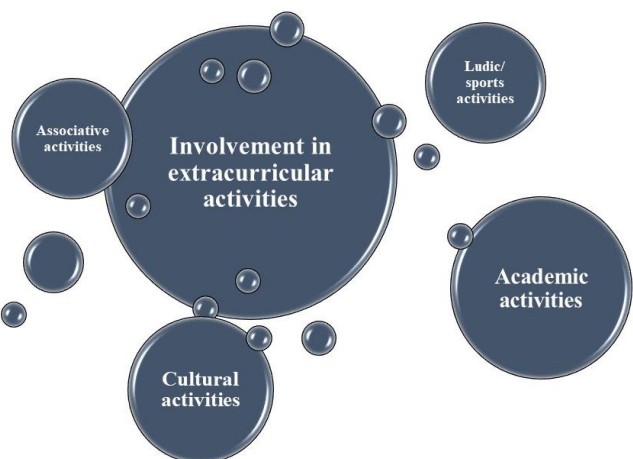

**Figure 1.** Students' involvement in extracurricular activities. Source: The author's production.

*3.2. The Institutional View*

In line with the students' perceptions, institutional leaders understand the student's path as multifaceted, extending beyond the academic or curricular component and viewing students' participation in extracurricular initiatives as relevant to their overall education. Therefore, HEIs encourage the participation of their students in voluntary, cultural, recreational, and sports activities, among others. However, the analysis of the institutional leaders' narratives points towards the perception of a low level of involvement and participation of students in the extracurricular activities offered by the institution and academic groups. The argument that students have a very heavy academic load, which does not allow them to set aside some of their time to participate in this type of activity, is used by some of these leaders, who recognize the need to "[ . . . ] have a more active raw material, but I also recognize that time is not enough for everything" (Vice-President). Furthermore, these leaders also refer to the Bologna Process as "[ . . . ] taking much more of students' time, often preventing them from taking a more active part in these activities" (Vice-President). However, this justification is not consensual, and it is even refuted by other leaders, who claim that students' academic loads are not, currently, "[ . . . ] different from it was 30 or 40 years ago" (Pro-Rector) and that "[ . . . ] the fact that students claim that they have a very intense academic load is no reason not to be present" (President of the Student Union). The institutional leaders' vision regarding the involvement of students in non-curricular activities is that sometimes it is difficult to attract them.

In addition to the possible time scarcity that leads students to not participate regularly in extracurricular activities, institutional leaders put forward other reasons for this behavior, specifically: (i) the personalities of those responsible for boosting such events; (ii) the current social trend for the reduced demand for cultural goods; (iii) certain scientific areas seen as more absorbing in terms of time devoted to study; (iv) the physical distance between the place where these activities take place and the residence of some of the students; (v) the reconfigurations in terms of HE students' profiles; and (vi) the new and different ways in which students relate to involvement in extracurricular activities, compared to students from previous generations. While acknowledging the existence of the insufficient adhesion of the students to these initiatives, some of the institutional leaders mention that, in certain kinds of actions, this insufficiency does not occur. Given that HEIs have a wide offer in terms of activities, the understanding is that there are possibilities for all students' needs, tastes, and availability.

In sum, according to institutional leaders' perceptions, students' participation in extracurricular activities is generally low, falling very often short of expectations. However, in some scientific areas that interest students, or given the possibility of personal and overall development, these actors end up engaging in actions that, while not being part of the courses' syllabus, are envisaged as educational opportunities offered by their institution.

## 4. Conclusions

The analysis of the interviewees' narratives allows for the conclusion that students' academic experience cannot be disconnected from their social and functional experience in the HEI. This stance confirms the importance, described in previous studies [22,24,26,30,33,37,50,74,75], of students' involvement in educational activities beyond the classroom boundaries. Corroborating literature, participants in this study posit that student life cannot be limited to class attendance and the preparation of papers and exams—this experience should be complemented with recreational or, at least, less formal activities. These can be materialized mainly in extracurricular activities, such as the participation in seminars, workshops, and conferences with perceived relevance to the enhancement of students' curriculum and as a complement to their education, but also in other activities, such as the participation in decision-making bodies and associative activity. Students perceive these activities as being a vehicle par excellence for the attainment and/or development of transversal competences, seen as an added value in their overall education, sometimes at the expense of one or two points in the final grade, which are seen as a small price to pay for the competences they have attained through their active participation in the life of the academy.

Most institutional actors who participated in this study believe, similarly to the students interviewed, that the students' involvement in extracurricular activities, i.e., more cultural or recreational activities, as well as their involvement in institutional decision-making bodies and associative movements, are privileged ways of complementing students' academic training and are perceived by them as pivotal in their overall education, both as professionals and as individuals. Furthermore, students' participation and involvement in decision-making bodies, such as the Pedagogic Councils or the General Councils, or in academic movements, such as student unions or student study groups, is understood as an excellent vehicle for students to attain competences that can be used later on, in the labor market. The issue of grades is even sidelined when compared to the attainment of transversal competences, especially by students involved in decision-making dynamics and bodies.

On the side of the HEIs, the involvement of students in institutional decision-making processes is also seen as central. Besides allowing students' involvement in making decisions that directly affect them, this is also a way to prepare them for their future professional lives through the attainment of participation and decision-making competences, among others. Students' engagement in extracurricular activities works, or should work, in conjunction and complementarity with the academic activities in the sense of students' overall education, while contributing to their degree of commitment to the HEI, their persistence in the institution, and, ultimately, their graduation. Thus, the results from this study, while revealing the relevance students ascribe to their integration in the HEI, namely through their engagement in extracurricular activities of a diverse nature, may assist institutional policymakers in defining and improving strategies for their students' integration and development.

This study is not without limitations. The first is due to the methodological approach chosen. The choice of a qualitative methodology prevents generalizations, inasmuch as the purpose of qualitative research is not to generalize the results but rather to describe, interpret, and ascribe meaning to the subjects' positioning concerning a given situation [76]. This is the main disadvantage of using qualitative approaches, as they do not allow the generalization of the results of studies with the same degree of certainty that quantitative approaches do. Moreover, the results of these studies cannot be tested to verify whether they are statistically significant or due to chance [77]. Moreover, qualitative research can provide answers to specific research questions that quantitative research cannot [78]. The second limitation concerns the number of cases studied and their geographic location. This study was limited to four case studies in northern Portugal, albeit the choice of the HEIs to be studied tried to obey some criteria that could make these cases, in some way, "representatives" of the Portuguese reality—by selecting a classic university, a new university, a large polytechnic institute, and a small polytechnic institute. The third limitation is that this

study encompassed only the public sub-sector of Portuguese HE, so it remains to analyze the reality of private HE. We believe that, while some of the dimensions of analysis used in this study would deliver similar results, others would possibly show relevant differences.

**Funding:** This research was funded by FCT—FOUNDATION FOR SCIENCE AND TECHNOLOGY, I.P., grant number UIDB/00757/2020 of CIPES—Centre for Research in Higher Education Policies, by national funds through FCT/MEC. The APC was funded by *Education Sciences*.

**Institutional Review Board Statement:** The study was conducted in accordance with the Declaration of Helsinki, and approved by the Review Board of CIPES–Centre for Research in Higher Education Policies (protocol code 05/2022 of 28 December 2022).

**Informed Consent Statement:** Informed consent was obtained from all subjects involved in the study.

**Data Availability Statement:** The data presented in this study are available on request from the corresponding author. The data are not publicly available due to privacy restrictions.

**Conflicts of Interest:** The author declares no conflict of interest.

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
