# Peer review of "Student Academic and Social Engagement in the Life of the Academy—A Lever for Retention and Persistence in Higher Education"

_education, doi:10.3390/educsci13030269_

Round 1

Reviewer 1 Report

TITLE: Academic and social engagement of students in the life of the academy. A lever for retention and persistence in higher education.

ABSTRACT: The abstract should be structured as IMRD (Introduction, Materials and Method, Results, Discussion) Here we only contextualise and give results not supported by the research conducted.

INTRODUCTION. Adequate contextualisation is given, but no research questions are posed and the research problem is not well defined.

THEORETICAL FRAMEWORK. It is adequately developed, although it relies heavily on Astin's theory of participation and more recent literature and studies should be sought.

METHODS. The objectives of the study are not clear.

The methodology chosen for this research study is qualitative, and multiple case study methodology was used. This leads to a sample of 58 academic actors, including students and institutional leaders from four Portuguese HEIs. A possible limitation of this study is the generalisability of the results, which is limited. It is not explained how the sample of students was selected, why undergraduate and postgraduate students were selected, and why the number was chosen. It is argued that heterogeneity was sought to be reflected, but it is not clear to what extent.

Semi-structured interviews are proposed, but it is not explained how the script for such interviews was generated.

RESULTS. For the analysis of results, a set of categories coded a priori from the critical review of the literature on the dimensions and factors that influence the academic and social experience of the students is mentioned, but these categories are not made explicit and are not clear after the Theoretical Framework provided. This should be clarified

The analysis of the students' narratives on participation in extra-curricular activities later on, four subcategories were identified: (1) participation in academic activities; (2) participation in cultural activities; (3) participation in recreational/sporting activities; and (4) participation in associative activities. The presentation of the data is done on the basis of these subcategories, and it is not clear how it relates to the categories announced above.

Perhaps a table linking categories and subcategories would help to better understand the data analysis.

Data from institutional leaders are provided, but there is no triangulation of results, which was announced in the section on the analysis of results where it says: The data collection instruments are three: (1) semi-structured interview with students; (2) semi-structured interview with institutional leaders; and (3) analysis of documents, to allow the triangulation of data. This point should be revisited

 DISCUSSION AND CONCLUSIONS: There is no authored discussion of results, this should be corrected.

BIBLIOGRAPHICAL REFERENCES. The bibliography consulted must be updated.

Author Response

Dear Reviewer,

Thank you very much for taking the time to review the manuscript and for the valuable comments and suggestions for improving its quality. The positive stance you took in making them is much appreciated.

In the attached document, I provide a point-by-point report on the revisions I made following your recommendations.

Warm regards,

The author

Reviewer 2 Report

Thanks for opportunity to review manuscript entitled ‘‘Student academic and social engagement in the life of the academy. A lever for retention and persistence in higher education’’ for Educational Sciences journal. In a qualitative research design, authors examined he benefits of academic and social engagement of students to their persistence and retention.  Up to my review, the merit of article is its novelty that rarely examined in literature. However, the article requires significant improvement almost all sections. Thus, some major and minor revisions required before publication of article. My main philosophy of reviewing a manuscript as reviewer and sometimes an editor to improve the manuscript and not punishing the authors, I provided very specific and detailed peer review of the manuscript to increase its quality and citation potential. If authors follow my suggestions, I think this article may be a publishable article. I hope authors of the manuscript may benefit from my review. Necessary and minor revisions reported section by section with the page and line number and when possible with suggestions. 

Title

1. Page 1, Line 1-2:  The title of the article must be corrected. One correct revision may be that ‘‘Student academic and social engagement in the life of the academy: A lever for retention and persistence in higher education’’

Abstract

2. Page 1, Line 14:   Authors must add information about method of the article including research design, participants, and data analysis process before following sentence ‘ ‘The results reveal that….’’

Introduction

3. Page 1, Line 28-31: The citation needed for following sentence ‘ ‘The major reconfigurations that the Higher Education (HE) arena worldwide has  been subject to over the past decades (in particular with the massification of this educational level, the increasing scarcity of resources, especially in financial terms, and the institutional competition for students), have caused a shift in the way higher education institutions (HEIs) relate with their pubic.’’ Moreover, please change pubic with another word it is not appropriate for a scientific article.

4. Page 1, Line 39: Please change mishap with more scientific word.

5. Page 2, Line 50-51: Please revise following sentence ‘ ‘There is a direct and positive relationship, well  documented in the literature, between students’ academic and social involvement and  their persistence and graduation’’ One revision maybe that ‘ ‘There is a well- documented direct and positive relationship between students’ academic and social involvement and  their persistence and graduation in the literature’’.

6. Page 2, Line 63-66: Please remove following no need these sentences ‘‘The paper is structured as follows: the next section offers a literature review on the  topic of HE student engagement in the life of the academy. Section 3 details the methodological choices that geared this study and section puts forth and discusses the results. Finally, section 5 offers the main conclusions of the research.’’

7. Manuscript, General: Authors must construct each paragraph at least with three sentences with maximum eight sentences as per APA 7 rules. Authors must correct this problem along the manuscript.

8. Manuscript, General: Authors must combine literature review section with Introduction section and must remove subtitle Literature review.

Method

9. Method, General: Author must construct Method design section using following subtitles ann must move all related information to related section. Research Design, Participants, Data collection Tools, Procedure, Data Analyses

10. Method, General: Author must add mean age with its standard deviation as wells as minimum and maximum age.

11. Method, General: Sample subtitle must change with Participants.

12. Method, General: Authors must add semi-structured interview questions for students and institutional leaders in a table.

13. Method, General: Authors must add sample categorization of themes in a table.

14. Method, General: Authors must give more detailed information about data analyses process in Data analyses section.

Findings and Discussion

15. Findings, General: The findings and discussion must be separate section. In this form , it is very difficult to separate findings from Discussion.

16. Findings, General:  Authors did not compare their findings with previous study findings in this section. In this form, Findings and Discussion section clearly just findings. Authors must compare their study findings with previous study divergent and convergent findings.

17. Limitations of study must be improved.

18. Practical implications of study is completely missing and must be added.

Author Response

(The authors gave the same response as above.)

Round 2

Reviewer 1 Report

After revision by the authors, the text has been substantially improved.

Reviewer 2 Report

Thanks for opportunity review revised manuscript entitled ‘‘Student academic and social engagement in the life of the academy. A lever for retention and persistence in higher education’’. I would like the thanks to authors. They make a good job for improving quality of their manuscript. Authors revised the manuscript as I requested with a good will. In this form, Introduction reflects  well the previous studies and study aim, Method section and Result section is correct, and Conclusion section adequately synthesis to previous study findings and current study results. Overall, I have no further comment regarding to manuscript. I congratulate to authors and wish them success on their future endeavors.